# Extracellular Vesicles from Human Cardiac Fibroblasts Modulate Calcium Cycling in Human Stem Cell-Derived Cardiomyocytes

**DOI:** 10.3390/cells11071171

**Published:** 2022-03-30

**Authors:** Brian X. Wang, Laura Nicastro, Liam Couch, Worrapong Kit-Anan, Barrett Downing, Kenneth T. MacLeod, Cesare M. Terracciano

**Affiliations:** National Heart and Lung Institute, Imperial College, London SW3 6LY, UK; l.nicastro19@imperial.ac.uk (L.N.); liamscouch@gmail.com (L.C.); worrapong.kit-anan12@alumni.imperial.ac.uk (W.K.-A.); b.downing@imperial.ac.uk (B.D.); k.t.macleod@imperial.ac.uk (K.T.M.)

**Keywords:** fibroblast, stem cell, cardiomyocyte, extracellular vesicles, isolation, ultrafiltration, chromatography, calcium cycling, excitation-contraction coupling

## Abstract

Cardiac fibroblasts regulate the development of the adult cardiomyocyte phenotype and cardiac remodeling in disease. We investigate the role that cardiac fibroblasts-secreted extracellular vesicles (EVs) have in the modulation of cardiomyocyte Ca^2+^ cycling–a fundamental mechanism in cardiomyocyte function universally altered during disease. EVs collected from cultured human cardiac ventricular fibroblasts were purified by centrifugation, ultrafiltration and size-exclusion chromatography. The presence of EVs and EV markers were identified by dot blot analysis and electron microscopy. Fibroblast-conditioned media contains liposomal particles with a characteristic EV phenotype. EV markers CD9, CD63 and CD81 were highly expressed in chromatography fractions that elute earlier (Fractions 1–15), with most soluble contaminating proteins in the later fractions collected (Fractions 16–30). Human induced pluripotent stem cell-derived cardiomyocytes (hiPSC-CMs) were treated with fibroblast-secreted EVs and intracellular Ca^2+^ transients were analyzed. Fibroblast-secreted EVs abbreviate the Ca^2+^ transient time to peak and time to 50% decay versus serum-free controls. Thus, EVs from human cardiac fibroblasts represent a novel mediator of human fibroblast-cardiomyocyte interaction, increasing the efficiency of hiPSC-CM Ca^2+^ handling.

## 1. Introduction

Cardiac fibroblasts play a critical role in the development and maintenance of the healthy adult cardiomyocyte. Cardiac fibroblasts account for 20% of the non-myocyte component of the heart and accumulate pathologically in disease [1]. They form extensive interactions with cardiomyocytes and are key regulators of the extracellular matrix. Due to their plasticity, during development and in disease, there is transition of quiescent fibroblasts into a more active and reversible phenotype, known as myofibroblasts, with contractile and tissue repair properties [1]. We have previously used animal models to show that cardiac fibroblasts are key regulators of cardiomyocyte function [2].

A major focus of the current research is the development and use of human induced pluripotent stem cell-derived cardiomyocytes (hiPSC-CMs), which have shown great promise as a replacement for adult human cardiomyocytes in cell therapy, drug discovery and in disease modelling. hiPSC-CMs display an immature phenotype more akin to neonatal or diseased cardiomyocytes. Notably, hiPSC-CMs have inefficient excitation-contraction coupling, marked by slow Ca^2+^ cycling kinetics [3,4,5]. Their naivety to the hormonal and mechanical myocardial environment, along with their stability in culture and ability to form a syncytium of cardiomyocytes in vitro, makes them an ideal model for the study of fibroblast-cardiomyocyte interactions.

Our group has recently shown that cardiac fibroblasts are important regulators of hiPSC-CM Ca^2+^ cycling, improving the efficiency of hiPSC-CM excitation-contraction coupling by recruiting the sarcoplasmic reticulum Ca^2+^ stores to Ca^2+^ cycling [6]. The mechanisms involved in fibroblast-mediated regulation of cardiomyocyte Ca^2+^ cycling, universally altered in disease, are still yet to be fully delineated.

In the past decade, extracellular vesicles (EVs) have emerged as a potential mediator of intercellular communication. EVs are often present in low quantities in cell culture media or biofluids, and potential soluble contaminating proteins are present in relatively high quantities [7,8]. The current gold standard for EV isolation is multistep differential ultracentrifugation combined with density-gradient ultrafiltration [9,10].

In this study, we used this technique to isolate EVs from biological samples. We then aimed to investigate fibroblast-cardiomyocyte interaction through the effect of fibroblast-secreted EVs on hiPSC-CM Ca^2+^ cycling, a key determinant of cardiomyocyte activity.

## 2. Materials and Methods

### 2.1. Cardiac Fibroblast Isolation and Culture

Human ventricular fibroblasts were obtained from explanted hearts of patients with dilated or hypertrophic cardiomyopathy at Harefield Hospital. Tissue samples from Harefield Hospital were provided by the Cardiovascular Research Centre Biobank at the Royal Brompton and Harefield NHS Foundation Trust, UK (NRES ethics number for biobank samples: 09/H0504/104+5; Biobank approval number: NP001- 06-2015 & MED_CT_17_079). The inferior 1/3rd of the heart was received.

A culture-based approach was used to isolate cardiac fibroblasts. Tissue culture-treated Petri dishes were coated with 10 µg/mL fibronectin in Phosphate-Buffered Saline (PBS) at 37 °C for 1 h during explant isolation and sterilization. Left ventricular free wall samples were collected in cold cardioplegia. Tissue pieces were washed and maintained in fresh sterile PBS containing 5% *v*/*v* Penicillin-Streptomycin (PBS-PS) while they were cut into small pieces (<1 mm^3^). The PBS-PS was aspirated, and the samples were maintained in 0.05% Trypsin-Ethylenediaminetetraacetic acid (EDTA) for 2 min. Trypsin was quenched with the same volume of fibroblast culture media (Dulbecco’s Modified Eagle Medium (DMEM), 10% *v*/*v* Foetal Bovine Serum (FBS), 1% *v*/*v* Penicillin-Streptomycin). Fibronectin solution in the prepared Petri dishes was replaced with 0.5 mL standard fibroblast culture media. Tissue pieces were arranged in the dishes with a distance of at least 5 mm between pieces to allow for outgrowth. These were incubated at 37 °C for 2 h before explants were submerged with additional standard fibroblast culture media. The cultures were incubated (37 °C, 5% CO_2_) for 4 days before the supernatant was changed for new fibroblast culture media. The explant supernatant was then changed every two days. Fibroblasts proliferating in the explant dishes were considered passage 0. At confluence, fibroblasts were trypsinised with 0.05% Trypsin-EDTA and incubated for 7 min (37 °C), before the fibroblast suspension was transferred to a tissue culture flask. Trypsin was quenched with the same volume of standard fibroblast culture media.

### 2.2. GCamp6f Stem Cell Dissociation and Maintenance

HiPSC line Wild Type C (WTC) GCamp6f hiPSC-CMs (Professor Bruce Conklin, Gladstone Institute) were maintained in tissue culture dishes coated overnight with 0.125% *v*/*v* Matrigel (Corning, Corning, NY, USA) diluted in Knockout DMEM (Life Technologies, Carlsbad, CA, USA). Dissociation was performed when the cells reached 80% confluence. Cells were washed with PBS and then incubated with 0.5 mM EDTA in PBS at room temperature for 5 min. The solution was removed and replaced with Essential 8^TM^ Medium (Thermo Fisher Scientific, Waltham, MA, USA) supplemented with 10 µM ROCK inhibitor (Stem Cell Technologies, Vancouver, BC, Canada) to remove the cells from the tissue culture plate. The cells in suspension were replated on tissue culture dishes coated overnight with 10 µg/mL Matrigel diluted in Knockout DMEM. The cells were incubated in Essential 8^TM^ Medium supplemented with 10 µM ROCK inhibitor for the first 24 h after replating before being maintained in Essential 8^TM^ Medium replaced daily until cells reached 80% confluence. At this point (day 0), the stem cells were dissociated or underwent differentiation.

### 2.3. GCamp6f stem Cell Differentiation

The hiPSCs were induced to differentiate into cardiomyocytes by biphasic modulation of Wnt signaling [11]. On Day 0, cells were exposed to 8 µM Wnt-activator CHIR99021 (MERCK Millipore, Burlington, MA, USA) in RPMI-1640 media supplemented with B27 minus insulin (RB–INS) (Life Technologies, Carlsbad, CA, USA) for 24 h. On day 1, the culture media was replaced with fresh RB–INS and maintained incubated for 48 h. On day 3, the RB–INS was removed, and cells were then incubated in 2.5 µM Wnt-inhibitor C59 (Biorbyt) in RB–INS for 48 h. On Day 5, this was changed to fresh RB–INS, which was changed every two days, until areas of beating cells became evident (~Day 9–11).

### 2.4. GCamp6f hiPSC-CM Purification

Once populations of hiPSCs have undergone differentiation, the mixed population of hiPSC-CMs and non-cardiomyocytes underwent purification before replating. Once areas of beating cells became evident, metabolic selection by glucose depletion was used to purify the cell population as previously described [12]. During these 96 h, the cells were maintained in RPMI-1640 with no glucose, supplemented with B27. Following metabolic selection, cultures were maintained in RPMI-1640 supplemented with B27 (RB + INS) for 2 days before replating.

Tissue culture dishes were prepared with 10 µg/mL fibronectin in PBS-PS and incubated for at least 30 min to allow for fibronectin attachment to the tissue culture plate. During this time, cultures were washed in PBS then incubated in 1 mL of 1× TrypLE per tissue culture plate well for 7 min before agitated using a P1000 pipette and transferred to a fresh 50 mL centrifuge tube. 1 mL of RPMI-1640 + 10% FBS was added to the cells per 1 mL of TryPLE before centrifugation at 200G for 5 min. The supernatant was removed before resuspension of the cell pellet in hiPSC-CM plating media (RPMI-1640 media supplemented with B27, 10% *v*/*v* FBS, 10 μM ROCK inhibitor + 1% *v*/*v* Penicillin- Streptomycin) for the desired density of 2.5 million live cells per 2 mL of media. PBS in the tissue culture dishes incubated with fibronectin in PBS (10 µg/mL) was aspirated before 2.5 million live cells were added per well of a 6-well plate. 24 h following replating, the media was changed for RB + INS and 1% *v*/*v* Penicillin-Streptomycin until D25-26, and replaced every two days.

### 2.5. Plating of hiPSC-CMs on Glass Substrate

On D25-26, hiPSC-CMs were washed twice in PBS before being detached by 1 mL of TryPLE then incubated at 37 °C and 5% CO_2_ for 8 min. Following incubation, a P1000 pipette was used to dislodge cells from the surface and gently mix to form a single-cell suspension. The dissociation solution was quenched by adding 5 mL RPMI 1640 + 10% *v*/*v* FBS. Cell suspension was then centrifuged for 200 G for 5 min. The supernatant was removed, and the cell pellet was resuspended in hiPSC-CM plating media.

Cells were plated onto 35 mm dishes with a 7 mm diameter glass bottomed well at a density of 80,000 hiPSC-CMs per dish. Dishes were pre-coated with fibronectin in PBS (10 µg/mL) for 1 h before cell plating, and PBS was aspirated before cells were plated. After incubation at 37 °C for 24 h, the media was replaced with RB + INS and 1% *v*/*v* Penicillin-Streptomycin, which was replaced every two to three days. Cells were used for experiments five to seven days after plating.

### 2.6. Fibroblast EV-Conditioned Media Production and Processing

Hyperflasks (Corning) were used to grow cardiac fibroblasts in fibroblast culture media until at least 70% confluence was achieved. After washing with PBS, cell cultures were incubated for 48 h in exosome-depleted fibroblast maintenance media (DMEM, 2% *v*/*v* Exosome-depleted FBS (Gibco, Waltham, MA, USA), 1% *v*/*v* Penicillin-Streptomycin). Following conditioning of the media, the media was removed and filtered through a 0.45 µm bottle-top filter.

### 2.7. EV-Conditioned Media Purification by Ultrafiltration

Fibroblasts conditioned media was centrifuged at 300× *g* for 10 min and 2000× *g* for 20 min to remove cells and debris, respectively. Following centrifugation, fibroblast-conditioned media was filtered through a 0.45 µm PES membrane to remove any remaining contaminant, and then concentrated by ultrafiltration using an Amicon Ultra-15 Centrifugal Filter (Merck) with a 100 kDa MWCO (Molecular Weight Cut-Off) (~7 nm pore size), at 4000× *g*, 4 °C for 3 min. This was repeated, topping up with additional media between centrifugations until a final volume of 500 was reached. Each concentrator processed 500 mL of starting conditioned media.

### 2.8. Purification by SEC

To exchange the buffer before column packing, 60 mL of Sepharose CL-2B slurry was transferred to a wide, shallow container and the gel beads were allowed to settle. The solvent was aspirated off and the beads resuspended in 120 mL of particle-free 18.2 MΩ water. This was repeated, before resuspending the slurry and pouring the suspension into a 30 cm Econo-Column (Bio-Rad) with a 50 mL syringe barrel attached as a reservoir. The column was allowed to pack until a final depth of 28 cm.

The packed SEC column was opened, and the PBS was allowed to drain through until the top of the beads in the separation media was dry. Loading of the sample into the column was carried out by applying 500 µL of sample to the top of the column using a 1 mL manual pipette. The column spout was then opened so sample could enter the column. Fraction collection began at this point. As soon as the media had wholly entered the separation media, PBS was gently added to the top of the column into the column headspace. Once at least 3 cm of PBS was added on top of the separation media, a PBS reservoir was connected to the column. For each sample that was processed in the SEC column, 30 fractions of 1 mL were collected, and their particle concentration was analyzed on a Nanosight NS300 to determine peak fractions. The protein content of the fractions was also analyzed by microBCA assay to check for efficient separation of EV from soluble proteins.

### 2.9. Nanoparticle Tracking Analysis (NTA)

Particle concentration in the fractions was quantified by NTA. The concentration and size distribution of particles in solution was measured using a Nanosight NS300 with an sCMOS (Scientific Complimentary Metal-Oxide Semiconductor) camera module and a 532 nm diode laser module. Samples were diluted 10–1000-fold in particle-free water from a Select Fusion Milli-Q water purifier (Suez Water, Trevose, PA, USA) to a concentration within the range of 10^8^–10^9^ particles/mL, such that the number of particles in the field of view was below 200/image. Using NTA V3.0 software, three 60-s videos were recorded and analyzed per sample, with customized software parameters Camera Level 15 and Detection Threshold 5. The finite Track Length smoothing algorithm was disabled.

### 2.10. Protein Quantification

Standard and working reagents for the microBCA (Bicinchoninic Acid) protein assay kit were prepared as per the manufacturer’s instructions (Thermo Fisher Scientific). Briefly, the working reagent was made by mixing components A, B and C in a 25:24:1 ratio and used immediately. Both samples and BSA standards were added with working reagent in clear-bottomed 96-well plates (50 µL of sample + 150 µL working reagent/well). Adhesive plate seals (Thermo Fisher Scientific) covered the top of the wells to prevent evaporation, and then plates were mixed thoroughly on a plate shaker for 30 s before incubation at 37 °C for 2 h. Colour change was assessed by measuring absorbance at 562 nm wavelength light on a µQuantTM micro-plate reader (Bio-Tek Instruments Inc., Winooski, VT, USA). The average absorbance measured in the blank standard replicates was subtracted from the standard absorbance, and the concentrations of unknown samples were determined using a linear regression fitted to the standards.

### 2.11. Dot Blotting

A 0.45 µm nitrocellulose membrane (Bio-Rad 162-0117, Hercules, CA, USA) was submerged in Tris-buffered saline (20 mM tris, 150 mM NaCl; pH 7.6) (TBS, Bio-Rad 170-6435) for 10 min. The Bio-Dot apparatus (Bio-Rad 170-6545) was assembled around the wetted membrane.

The four screws were tightened to prevent sample contamination between wells. All wells were rehydrated with 200 µL TBS and allowed to drain. 50 µL samples were added to a 96-well plate before being transferred to the sample template using a multi-channel pipette. The sample was allowed to pass through under gravity. All wells were washed three times with TBS and allowed to drain. The BioDot was then disassembled before the blocking of the membrane. The membrane was cut for orientation, marked with pencil for identification and then incubated in a solution of 5% *w*/*v* BSA in TBS-T (TBS-Tween) (20 mM tris, 150 mM NaCl, 0.1% *v*/*v* tween 20; pH 7.6) for 1 h at room temperature. The dot blot membrane was then washed three times in TBS-T for 10 min each round at room temperature before incubation overnight at 4 °C with the primary antibody in the desired dilution (Table 1) in TBS-T.

Following the overnight incubation, membranes were washed three times in TBS-T for 10 min each at room temperature, before incubation for 1 h at room temperature with the secondary antibody in the desired dilution (1:15,000) in TBS-T. The secondary antibody used was goat anti-mouse 800 CW (LI-COR 926-32210) Membranes were then washed three times in TBS-T for 10 min at room temperature and imaged using an LI-COR Odyssey imager.

Dot blots were quantified using Image Studio Lite V5.2 software (LI-COR). For each image, the background was subtracted with a rolling ball radius set to 25 pixels before a rectangle was fitted around each lane and the profile plot was viewed. The intensity of each dot was measured using the gel analysis procedure. The integrated density for each fraction was then expressed as a percentage of the total of all the dots. 

### 2.12. Cryo-Electron Transmission Microscopy (CryoTEM)

Holey Carbon Grids (HC200-Cu, Electron Microscopy Sciences, Hatfield, PA, USA) were plasma treated (15 s, O_2_/H_2_) on a Gatan Solaris Plasma Cleaner. Samples of EVs in PBS for cryoTEM were prepared using a Leica EM GP automatic plunge freezer. 4 µL of sample was added onto the carbon coated side of the grid while in an environmental chamber (relative humidity: 90%, temperature: 20 °C). Excess sample was blotted on filter paper and the obtained film was vitrified in liquid ethane. Samples were then stored under liquid nitrogen until imaged. Samples were imaged using a JEOL 2100 plus with 200 kV and the Minimum Dose System. The imaging temperature was −170 °C in a Gatan914 cryoholder. Micrographs were taken using a Gatan Orius SC1000 camera at a magnification of 30 k.

### 2.13. Immunogold Labelling

Isolated EVs were fixed in 2% paraformaldehyde (*w*/*v*) for 60 min, where 10 µL of fixed EVs were placed on a formvar-coated grid for 60 min. Each grid was placed in three droplets of 0.5M sodium cacodylate buffer (pH 7.4) for 5 min, followed by placing on a droplet of 1% Bovine serum albumin (BSA) for 30 min. Grids were then placed on 10 µL of CD63 (abcam) at 1:10 dilution for 60 min. Unbound antibody was removed by floating the grids on three droplets of 1% BSA for 5 min each. Grids were then placed on gold conjugated goat anti- mouse IgG (10 nm) at 1:50 dilution for a further 60 min followed by 3 washes in PBS and a final wash in distilled water. Grids were stained with 2% Uranyl acetate for 3 min, washed in distilled water, and viewed in a JEOL 1200 EX transmission electron microscope. Digital micrographs were taken using a Gatan digital micrograph.

### 2.14. Treatment of hiPSC-CMs with EVs

EVs were concentrated before treatment of the hiPSC-CM monolayers in glass-bottomed tissue culture dishes. Particles in SEC fractions 7–11 in solution were combined with 7 mL serum-free media (M199, 1× ITS) and placed in a 100 kDa MWCO Amicon Ultra-15 Centrifugal Filter and ultra-filtrated in a centrifuge at 4000× *g*, 4 °C for 3 min. Serial ultrafiltration at varying durations was carried out until a concentration of 150 µg per 100 µL of the sample remained. One hyperflask containing approximately 25 million fibroblasts produced approximately 300 µg of protein in the 5 mL of solution eluted in SEC fractions 7–11. From an original starting volume of approximately 250 mL of fibroblast-conditioned media, 150 µg of protein was added to the hiPSC-CMs.

To investigate the contribution of the non-EV content of the exosome-depleted serum-containing fibroblast maintenance media, the same starting volume of the exosome-depleted fibroblast maintenance media (500 mL) was processed in the same manner as the fibroblast-conditioned media. The articles in SEC fractions 7–11 were concentrated to a volume of 200 µL. 100 µL was added to the hiPSC-CM monolayer.

The effects of exosome-depleted fibroblast maintenance media with and without fibroblast-conditioned EV content were compared to serum-free media.

### 2.15. Ca^2+^ Transient Measurements

GCamp6f hiPSC-CMs were excited using a 470 nm wavelength light-emitting diode (LED) and emitted fluorescence collected through a 530 ± 35 nm long-pass filter. All Ca^2+^ transient recordings were carried out in standard Tyrode’s solution (140 mM NaCl, 4.5 mM KCl, 10 mM glucose, 10 mM HEPES, 1 mM MgCl_2_, 1 mM CaCl_2_; pH 7.4) at 37 °C. The cells were field stimulated at 1 Hz using a 20 ± 10 V pulse of 5 ms duration, and recordings were captured using a NeuroCMOS camera (Redshirt Imaging, Decatur, GA, USA) at 0.5 kHz with a temporal bin of 2 (final frame rate 250/s) using a 40x oil-immersion objective. Recordings were made after the cells had reached a steady state under field stimulation. The first four transients of each recording were signal-averaged before the following parameters were calculated using custom in-house software for MATLAB R2019b (MathWorks, Cambridge, UK): normalized fluorescent amplitude (F/F_0_, where F is peak fluorescence intensity and F_0_ is baseline fluorescence), time to peak (calculated as time from stimulus to peak fluorescence), and time to 50% and 80% decay (calculated as time from peak fluorescence to 50% and 80% reduction in amplitude, respectively). Inclusion criteria is the ability to field-stimulate the hiPSC-CMs at 1 Hz. Based on this, one, three and five preparations of control, EV-depleted serum and fibroblast-EVs conditions, respectively, were excluded in the Ca^2+^ cycling studies.

### 2.16. Statistical Analysis

All statistical analysis was performed using GraphPad Prism version 8 for Windows (GraphPad Software, San Diego, CA, USA). All data were subjected to the D’Agostino & Pearson omnibus normality test. For parametric distribution, when comparing the means of two sample groups, a two-way paired *t*-test was used. For comparisons of three or more sample groups, where observations were linked, a one-way within-subject ANOVA (Analysis of Variance) was used. Post-hoc tests used with ANOVA were either Tukey’s test (for comparison of all means) or Dunnett’s (for comparison to a single control value). For non-parametric distribution, a statistical test was performed using Mann-Whitney (2 sample groups) or Krustal-Wallis (Three or more sample groups). Unless otherwise stated, the data is expressed as the mean ± standard error of the mean (SEM), *n* = number of technical replicates in each group, and *N* = number of biological replicates in each group. To account for the biological variability of hiPSC-CMs, four batches of hiPSC-CMs were used for micro-BCA protein assays and Ca^2+^ transient cycling. Statistical significance is highlighted by * *p*-value < 0.05; ** *p*-value < 0.01; *** *p*-value < 0.001; **** *p*-value < 0.0001.

## 3. Results

### 3.1. Purification of Fibroblast-Conditioned Media

Human cardiac fibroblasts were collected from the ventricular wall of dilated cardiomyopathy patients and cultured in standard fibroblast culture media until passage 4–6 and characterised in Figure A1. Fibroblasts after passage 4 had high expression of alpha-smooth muscle actin (Figure A1 in Appendix A), therefore the effects studied here refer to activated fibroblasts or myofibroblasts. Quiescent, alpha smooth muscle actin negative fibroblasts were not studied.

The cultures were maintained for 48 h in exosome-depleted fibroblast maintenance media before the supernatant was collected, centrifuged to remove cells and debris, and filtered to remove large contaminating particles (>450 nm diameter) (Figure 1A). SEC then separated the remaining particles into 30 separate 1 mL fractions (Figure 1A). Micro-BCA assay of the SEC fractions demonstrated a protein profile with two distinct peaks. One smaller peak in protein content, thought to be the EVs, was consistently found at fraction 8 and a much larger peak composed of soluble contaminating proteins eluted at fraction 21 (Figure 1B,C).

We carried out NTA to analyse the size distribution of particles in the SEC fractions 5–13 (Figure 1B,C) (see Figure A2 in Appendix A and Table A1 in Appendix B). Plotting the two values against each other for fractions 7–12 from the SEC elution shows that the two values are closely correlated (r^2^ = 0.9581) (Figure 1D). Fractions 7–12 show particle content with a modal distribution between 50–150 nm, which is within the expected range for EVs (Figure 1E).

### 3.2. Purification and Characterisation of EVs from Human Cardiac Fibroblasts

The morphology of particles eluting from the column were assessed by cryo-TEM to identify the presence of cardiac fibroblast EVs (Figure 2A). Cryo-TEM imaging of fibroblast-conditioned media shows liposome-like particles with characteristic EV morphology between 50–200 nm diameter. Immunogold-EM identified that the particles had high expression of EV marker CD63 (Figure 2B). Dot blot analysis of the fibroblast-conditioned media probing for EV markers CD9, CD63 and CD81 identified the highest signal intensity for CD9 and CD81 in SEC fraction 8 (Figure 2C). This fraction was shown by the NTA and microBCA to have the highest overall particle and protein content, respectively (Figure 1C).

### 3.3. Fibroblast EVs Abbreviate hiPSC-CM Ca^2+^ Transients

Having determined that fractions 7–11 were enriched with fibroblast EVs, 150 µg of these were applied to hiPSC-CM monolayers for 24 h to assess the effect on hiPSC-CM Ca^2+^ cycling. We measured Ca^2+^ transient amplitude; Ca^2+^ transient time to peak and time to 50% and 80% decay (Figure 3A) [13].

Ca^2+^ transient duration was abbreviated by fibroblast EVs (Figure 3B), indicating an increase in Ca^2+^ cycling efficiency. This was largely due to a reduction in time to peak (Figure 3C), an indicator of Ca^2+^ release efficiency. There was no change in amplitude caused by EV-depleted fibroblast maintenance media in the presence or absence of fibroblast-conditioning (Figure 3D). Time from the Ca^2+^ transient peak to 50% decay for hiPSC-CMs was significantly reduced after fibroblast EV treatment versus serum-free control, but not EV-depleted serum versus control (Figure 3E, *p* = 0.16), with no difference to 80% decay (Figure 3F).

## 4. Discussion

Fibroblasts represent a key mediator of cardiomyocyte function within the heart. In their highly secretory form, they are thought to drive the maturation of cardiomyocytes in the developing myocardium into the adult phenotype as well as trigger the pathological changes that occur in disease. However, the mechanisms involved in fibroblast-mediated modulation of cardiomyocyte Ca^2+^ cycling are unclear. We show here that EVs from human cardiac fibroblasts improve the efficiency of hiPSC-CM Ca^2+^ handling and represent a novel mediator of fibroblast-cardiomyocyte interaction.

### 4.1. Effects of Cardiac Fibroblast EVs on Ca^2+^ Cycling

We identified that fibroblast-secreted EVs significantly abbreviate hiPSC-CM Ca^2+^ transients through a reduction in time to peak and time to 50% decay. This is in line with previous studies that have shown that cardiac fibroblast co-cultured with hiPSC-CMs and adult rat cardiomyocytes can increase the efficiency of Ca^2+^ cycling [2,6]. EVs are known to deliver a plethora of cargo content [14,15], and this study shows that the fibroblast EVs deliver cargo important in the modulation of Ca^2+^ cycling, although further studies are required to identify the bioactive components. However, it is important to consider that exogenous stimulation of cells is known to alter their phenotype, including their EV secretome [15]. Protein expression of fibroblasts changes in vitro, triggered by the differences in the biochemical, electrical and mechanical stimuli between the native myocardium and the in vitro environment. We identified an increase in expression of alpha-smooth muscle actin, a protein expressed upon differentiation of fibroblasts into the more active, diseased phenotype–myofibroblast (Figure A1). As it is now established that myofibroblasts are an activated, reversible fibroblast phenotype with greater secretory activity [1], here we use the term fibroblast to refer to the activated phenotype we observed. Therefore, our results are applicable to the dynamic regulation between cardiomyocytes and activated fibroblasts during development and disease and suggest that there is an abbreviation of hiPSC-CM Ca^2+^ cycling in this context. Further studies will require investigation into the role of fibroblasts in the quiescent state, which may be important in normal physiology when activation of fibroblasts is low.

It is important to also comment on the limitations to which an in vitro study can recapture the environment of the native myocardium. During development and in cardiac disease, cardiac fibroblasts proliferate rapidly and are stimulated by the biochemical, electrical and mechanical triggers. We speculate that this brings about the secretion of EVs that act on the neighbouring cardiomyocytes. It is likely that the activity of fibroblasts is graded in response to the severity of changes within its environment and the triggers from other non-myocyte components of the myocardium. It is possible that the proximity between fibroblasts and cardiomyocytes allows the cardiomyocytes to experience a high concentration of EVs which is difficult to reproduce in the in vitro environment. In our study, we culture the hiPSC-CMs in a relatively high dose of EVs. Although we are unable to show that this is the same amount of EVs as experienced in the disease myocardium, it does demonstrate that fibroblast-secreted EVs are a modality by which cardiac fibroblasts modulate hiPSC-CM Ca^2+^ cycling.

### 4.2. EV Isolation

When considering EV isolation techniques, the choice of isolation method impacts the purity and yield of collected EV samples [7,10,15,16]. Prior to 2015, ultracentrifugation was used in 90% of studies with EV isolation [10]. Ultracentrifugation can be further subcategorized into three different variations with approximately equal distribution: regular ultracentrifugation, ultracentrifugation with a washing step, and density-gradient ultrafiltration. Ultrafiltration with and without a washing step produced an equal yield, higher than the more labour-intensive procedure of density-gradient ultrafiltration, but has been criticized for producing EV samples that have high contamination from soluble content [9,10]. Furthermore, these procedures also often cause extensive damage to the EVs and demonstrates very low reproducibility compared to other emerging techniques [6,17,18].

In this study, we used a method based on the combination of centrifugation and ultrafiltration followed by SEC. Based on Webber and Clayton’s proposed integration of quantification methods to assess the purity of EV samples, the ratio of particle concentration to protein concentration was detected as 4.67 × 10^9^ particles/μg of protein, indicating an intermediate purity (Figure 3) [17]. It is important to consider that this ratio may vary between cell lines. As cardiac fibroblasts are known to be highly secretory cell types, they are likely to produce many proteins of similar size to the EVs. Combining our size-based purification technique with other techniques for EV purification could increase the purity. However, this does provide a useful reference point.

In our study, we pool the SEC fractions with high EV content to investigate the role that fibroblast-secreted EVs have on hiPSC-CM Ca^2+^ cycling. As we have now shown that EVs do play a significant role in hiPSC-CM function, future studies should investigate the EV content that is important in mediating the intercellular crosstalk, and how the content may change across EVs of different sizes.

### 4.3. Cardiac Fibroblast EVs and Their Role in Paracrine Signaling

We investigated the effects of EVs secreted from cultured human cardiac fibroblasts. These fibroblasts were cultured in fibroblast maintenance media containing standard FBS, a blood product with very high concentration of bovine EVs [19]. Fibroblasts are heavily dependent on FBS-content, and omission of FBS could change EV composition and cell survival, so collecting fibroblast-secreted EVs in media void of FBS risks substantial contamination with apoptotic bodies and other cellular debris. Comparing the protein yields from studies that use EV-depleted serum and studies that abruptly changed culture media to serum-free conditions showed that studies using EV-depleted serum reported a significantly higher average yield than the serum-free conditions [10]. For the purposes of this study, we chose to limit the EV collection time to 48 h and use EV-depleted FBS during the collection period [20]. Future studies should compare the secretome of cardiac fibroblasts maintained in serum free media and EV-depleted serum-containing media.

It has been widely agreed that cardiac fibroblasts are important regulators of Ca^2+^ cycling in hiPSC-CMs and are the native cardiomyocyte in physiology and in disease. Our group recently showed that direct contact co-culture causes an abbreviation in Ca^2+^ cycling, but co-cultures without direct contact between the two cell types, and treatment with cardiac fibroblast-conditioned media, had no effect on Ca^2+^ decline and prolonged the time to peak [6]. Building on this understanding, we postulate from our results in the current study that soluble mediators in the fibroblast-conditioned media, including cell debris and soluble, non-EV factors released by the active fibroblastic phenotype in vitro have divergent effects on CMs versus fibroblast-secreted EVs. A higher localized concentration of EVs in the intercellular space between the two cell types may also play a role in mediating the abbreviation of hiPSC-CM Ca^2+^ transients identified in co-cultures.

## 5. Conclusions

We validated an effective EV isolation and purification protocol that produces EVs with bioactive properties. We identified that fibroblast EVs are involved in modulating hiPSC-CM Ca^2+^ cycling in vitro and represent a novel mediator of human fibroblast-cardiomyocyte interaction.

## Figures and Tables

**Figure 1 cells-11-01171-f001:**
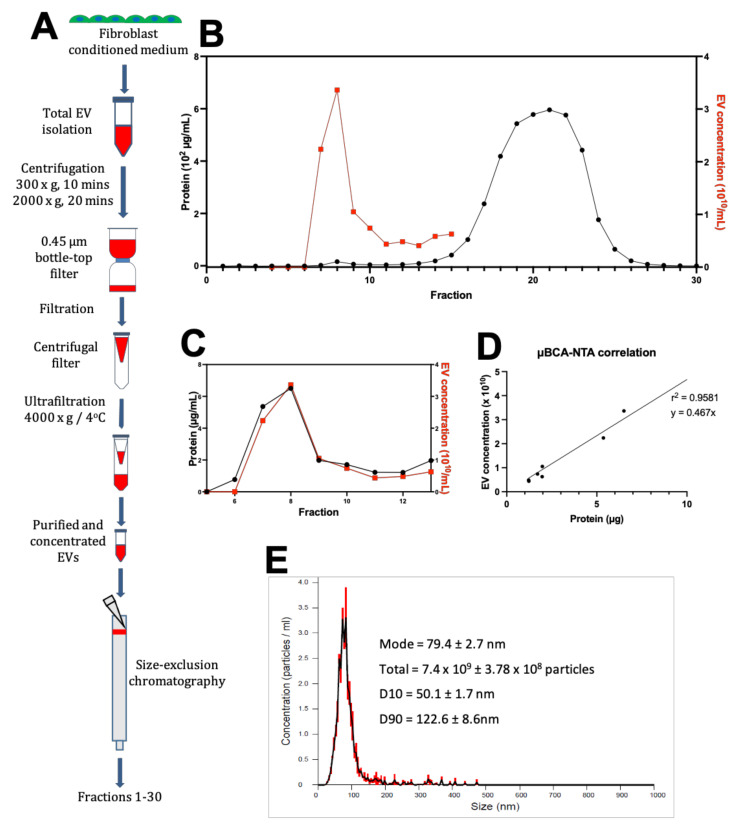
Isolation and purification of human fibroblast extracellular vesicles (EVs). (**A**) EVs were isolated from cultured human cardiac fibroblasts by a series of centrifugation followed by filtration and ultrafiltration at 4000 g until the equivalent of 500 mL of fibroblast-conditioned media was concentrated to 500 µL. 250 µL of concentrated media was run through the chromatography column and thirty 1 mL fractions were collected. (**B**) Elution profile from size-exclusion chromatography shows micro-BCA assay protein concentration (black) (*N* = 4, *n* = 4) against nanosight tracking analysis (NTA)-measured particle concentration in fractions 4–15 (red) (*N* = 1, *n* = 1). (**C**) Protein and EV distribution of SEC fractions 5–13. (**D**) Correlation between particle concentration and protein content in SEC fraction 7–12. Straight line of best fit identifies the presence of 4.67 × 10^9^ particles/μg of protein. (**E**) Representative NTA detection of particles in SEC fraction 10 with a 1:25 dilution. Concentration detected is in ×10^7^ particles/mL. Total particle count was calculated as NTA particle count multiplied by 25 to account for the dilution.

**Figure 2 cells-11-01171-f002:**
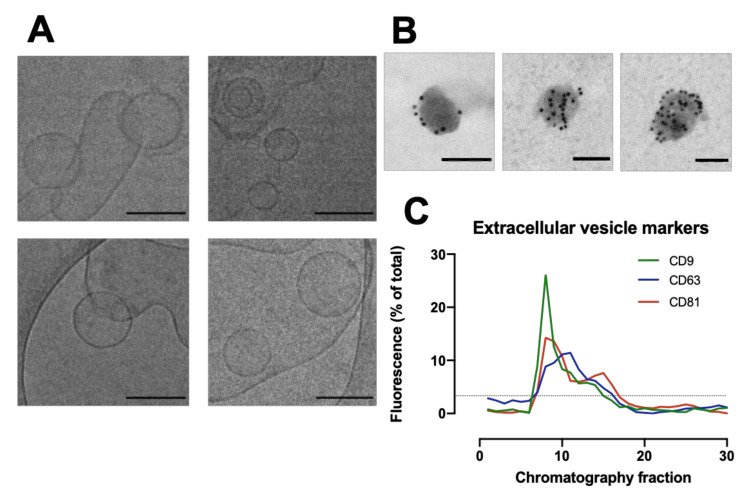
Characterisation of extracellular vesicles (EVs) from human cardiac fibroblasts. (**A**) Cryogenic Transmission electron microscopy (cryo-TEM) of fibroblast-conditioned media. Scale bar = 100 nm. (**B**) Immunogold-electron microscopy tagging for CD63 identified lipophilic particles with high CD63 expression. Scale bar = 100 nm. (**C**) Quantification of dot blots of fibroblast-secreted EV samples against CD9, CD63 and CD81 as percentage of total signal. Dotted line represents mean (3.33). *N* = 2 preparations.

**Figure 3 cells-11-01171-f003:**
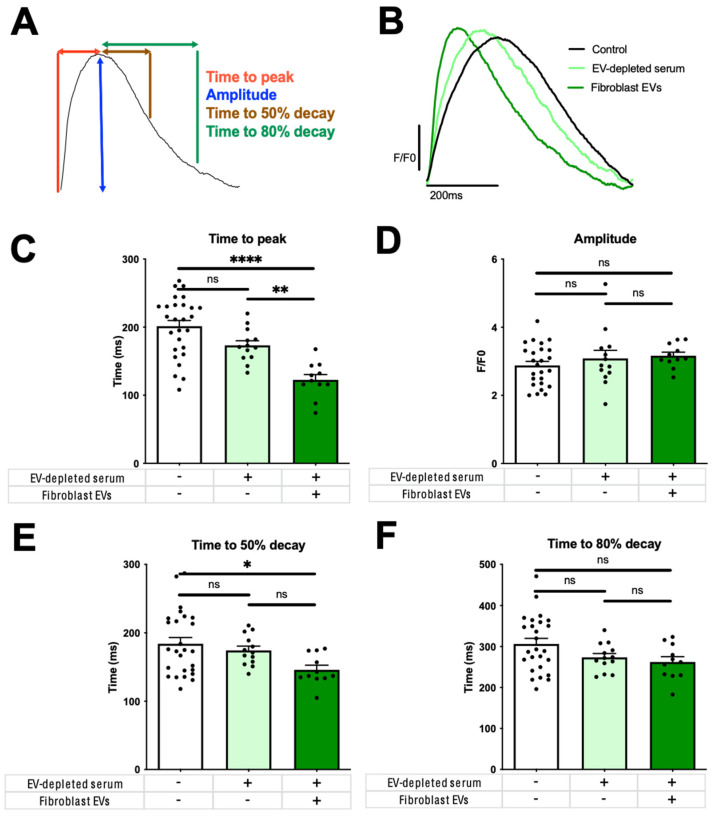
Modulation of human cardiomyocyte Ca^2+^ cycling by fibroblast-secreted extracellular vesicles (EVs). (**A**) Human induced pluripotent stem cell-derived cardiomyocyte (hiPSC-CM) Ca^2+^ transient parameters measured were time to Ca^2+^ transient peak as an indicator of Ca^2+^-induced Ca^2+^-release efficiency; amplitude as an indicator of cytosolic Ca^2+^ availability, and time from peak to 50% and 80% decay as an indicator of the efficiency of cytosolic Ca^2+^ removal mechanisms. hiPSC-CMs were field-stimulated at 1 Hz. (**B**) Representative traces of control (serum-free media) (black line in **B** or white bar in **C**–**F**), EV-depleted fibroblast maintenance media (Dulbecco’s Modified Eagle Medium, 2% *v*/*v* exosome-depleted Foetal Bovine Serum, 1% *v*/*v* Penicillin-Streptomycin) in the absence of fibroblast EVs (light green) and conditioned with fibroblast EVs (150 μg) (dark green). Individual parameters were Ca^2+^ transient (**C**) time to peak; (**D**) amplitude and time from peak to (**E**) 50% decay and (**F**) 80% decay. Error bars represent SEM. Control *N* = 7, EV-depleted serum *N* = 4, Fibroblast EVs *N* = 4 biological replicates of hiPSC-CMs. Fibroblast EVs taken from four diseased human hearts. *n* = 4 technical replications. One, three and five preparations of control, EV-depleted serum and fibroblast-EVs conditions, respectively, were not suitable for experiments due to human or technical error. * = *p* < 0.05, ** = *p* < 0.01, **** = *p* < 0.0001.

**Table 1 cells-11-01171-t001:** Primary antibodies for dot blot.

Primary Antibody	Dilution Factor	Species	Manufacturer	Product Code
Anti-CD9	800	Mouse	Thermo Fisher Scientific	10626D
Anti-CD63	400	Mouse	Thermo Fisher Scientific	10628D
Anti-CD81	400	Mouse	Thermo Fisher Scientific	10630D

## Data Availability

The data underlying this article will be shared on reasonable request to the corresponding author.

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
