# Peer review of "Extracellular Vesicles from Human Cardiac Fibroblasts Modulate Calcium Cycling in Human Stem Cell-Derived Cardiomyocytes"

_cells, 2022, doi:10.3390/cells11071171_

Round 1

Reviewer 1 Report

no further comments.

Author Response

We thank reviewer 1 for approving the manuscript in its current form.

Reviewer 2 Report

The major issue is the lack of plausible mechanism explaining how EV affect Ca cycling. Expression of protein/channels (SERCA2, NCX1, RyR2) as well as the phospholamban phosphorylation should be assessed to shed light on the mechanisms of calcium cycling.

In their previous work, frequently cited in the present manuscript (10.1016/j.jacc.2018.06.028.), Authors showed that fibroblast conditioned medium (very likely containing EV) had no effect on calcium cycling, while the EV affect significantly such parameter in the present manuscript. How do Authors explain this discrepancy?

It would be very interesting to test and compare the effect of proteins fraction (picking at fraction 20). Soluble fraction has also effect.

Reviewer 3 Report

In this paper Wang and co-workers investigated the role that cardiac fibroblasts-secreted EVs have in the modulation of cardiomyocyte Ca2+ cycling.

This is a very interesting study conducted by a group with experience in the area of stem cell-derived myocardium. The aim of the study is clear, the paper is well written, the surprising conclusions are adequately supported by the data included in the manuscript. I would propose to accept the manuscript in the present form.

Author Response

We thank reviewer 3 for approving the manuscript in its current form.

Round 2

Reviewer 2 Report

Authors did not perform new experiment, the paper has not been improved Since they stated that there are potential soluble factor that counteract EV effects it is in my opinion crucial to compare soluble factors (high fractions) vs EV.

This manuscript is a resubmission of an earlier submission. The following is a list of the peer review reports and author responses from that submission.

Round 1

Reviewer 1 Report

The MS by Wang et al. presents results showing that fibroblast EVs may interfere with cardiomyocyte calcium handling. The topic may be of interest for a wide audience, but the MS has several shortcomings.

  1. The overall aim of the MS is unclear. It is not stated either in the introduction or in the discussion whether the effect of EVs on Ca-handling or the effect of EV isolation on EV effect is aimed here. Although the experimental design suggests the former, texts lean towards the latter. In case the latter is aimed, comparative studies need to be included.
  2. Since fibroblasts are obtained from patients with cardiomyopathies, can they be considered as diseased cells? In this case healthy fibroblasts should also be used here. However, it also can be hypothesized that fibroblasts lost their diseased phenotype during the 2 weeks-long culture prior to the EV isolation. In this case, authors should evidence the healthy nature of the fibroblasts.
  3. It is unclear why the authors did not start EV isolation with low-speed centrifugation to remove cellular debris and lEVs, rather with a 200nm filtration which can shred larger particles and fragment EVs, therefore contaminate the isolate and distort EV composition. To exclude this possibility authors should e.g., show quantitative evaluation of EM imagery, evidencing the ratio of intact to damaged vesicles. It is also unclear why did not the authors elute SEC in serum-free media, thus avoiding the last session of centrifugation. If due to dilution during SEC, smaller colum could have been used, since sample volume up to 10% of column volume gives acceptable separation with significantly less dilution.
  4. The description of Ca-imaging method is missing.
  5. It is unclear why this concentration of EVs was used here. According to the methods section, the applied 150ug EV protein corresponds to the EV production of about 12.5 million fibroblasts for 48h. It seems to be a tremendously high dose of EVs from a number of fibroblasts that vastly exceeds that of the recipient myocytes. Besides justification and argument for in-vivo relevance, dose-response measurements are highly recommended. It is also unclear why the doses were not set based on the EV number instead.
  6. It is unclear how many replicates have been performed in which step. For example, it is not disclosed how many donor samples were used, how many independent cultures were maintained, how many isolates were produced from these cultures, how many individual hIPSCM cultures were treated in how many technical replicates, and how many action potentials on how many cells were recorded and averaged in these replicates. It is also unclear why the number of datapoints are this highly different in Fig 3 and if there is a biological justification behind it.
  7. Since Ca-imaging method is not described, it is unclear whether cardiomyocytes were beating spontaneously or paced. How can the authors exclude the possibility that the observed changes are due to a difference in the contraction frequency?
  8. The MS should provide some mechanistic evidence, via what mechanism EVs may influence Ca-handling. Is it receptor-mediated? Is it via induction of gene-expression, or a direct effect on ion channels?
  9. The end of the abstract is unclear, as if sentences are repeated.
  10. Fig2A is redundant to Fig1A.
  11. It is recommended to provide wide-field images for EM to better represent size distribution of EVs and to show purity of isolate, e.g., as a supplementary material.
  12. Language editing is recommended.

Reviewer 2 Report

The manuscript reports a clearly written and well-structured study investigating the function of fibroblast derived exosome on cardiomyocyte calcium homeostasis. Fibroblast and cardiomyocyte interaction might play important role in heart function and diseases. Thus this study provides a potential model how they interact with each, which might be very interesting to the CVD field.

My biggest concern is the purity of the exosome with SEC isolation. As the authors  stated in the discussion the purity is intermediate. It is known that SEC method might introduce some soluble contamination in the exosome preparation. More evidence to prove the purity of the exosome preparation are needed to support the final conclusion.

Further comments refer to the choice of cell line. Does this observation only happen with exosome from fibroblasts? or any exosome will do? In that case, a negative control cell line like HEK293 might be included as well.

Round 2

Reviewer 1 Report

Re 1: Section 4.2, EV isolation, is unrelated to the effect of fibroblast EVs on CMs, therefore, it is recommended to be removed.

Re 2: In the opinion of the reviewer it is not recommended to use terms fibroblast and myofibroblast interchangeably. Authors should clarify which condition they would like to model here. A heart undergoing remodelation where myofybroblasts are present? It would also be recommended to formulate P11l31-32 as a limitation, since this assumption is not supported by any experimental evidence here.

Re 5: The reviewer disagrees with the authors. It is possible to show physiological relevance of FB-EVs on CM Ca-handling, for example by co-culturing experiments or via transferring unfractionated supernatants from fibroblasts to CMs. Since exactly this kind of experiments were performed in the reference #5, authored by the same group, and since those experiments did not show any effect on Ca-handling of CMs, conclusions drawn here are not justified. One explanation of the results presented here may be that the used samples of the enormous amount of EVs had been derived from large volumes of supernatants and therefore they may have carried over and specifically concentrated certain non-EV materials. Even if another control group would be inserted here, in which CMs are treated with samples derived from naïve culture media (that had not been put on fibroblasts) treated the same way as the current EV preparation, it is very difficult to defend physiological relevance of the current paper in the light of ref#5..

Re 6: The justification of replicate numbers and exclusions should be included in the methods section. Please also detail what “media were not suitable for experiments” mean, what the measure and threshold was that you evaluated for this decision?

Re 11: Please align the MS with recommendations of MISEV2018 (DOI: 10.1080/20013078.2018.1535750). In Section 4-c it calls for wide-field EM. Please include such imagery.

Reviewer 2 Report

All my questions/concerns have been addressed. I think it is ready for publish.